# Trends in Health-Risk Behaviors among Chinese Adolescents

**DOI:** 10.3390/ijerph16111902

**Published:** 2019-05-29

**Authors:** Lan Guo, Tian Wang, Wanxin Wang, Guoliang Huang, Yan Xu, Ciyong Lu

**Affiliations:** 1Department of Medical statistics and Epidemiology, School of Public Health, Sun Yat-sen University, Guangzhou 510080, China; guolan3@mail.sysu.edu.cn (L.G.); wangt97@mail2.sysu.edu.cn (T.W.); wgg0808@163.com (W.W.); 2Guangdong Engineering Technology Research Center of Nutrition Translation, Guangzhou 510080, China; 3Center for ADR monitoring of Guangdong, Guangzhou 510080, China; hgl13538960848@163.com (G.H.); xuyan2@163.com (Y.X.)

**Keywords:** health-risk behavior, trend, adolescent, wave

## Abstract

Adolescent health-risk behaviors can have long lasting negative effects throughout an individual’s life, and cause a major economic and social burden to society. This study aimed to estimate the prevalence of the health-risk behaviors among Chinese adolescents and to test the trends in health-risk behaviors without and with adjusting for sociodemographic characteristics. Data were drawn from the School-based Chinese Adolescents Health Survey, which is an ongoing school-based study about the health-risk behaviors among Chinese adolescents (7th to 12th grade). During the first wave through the third wave, the prevalence of lifetime, past 12-month, and past 30-day use of opioid decreased by 4.19%, 0.63%, and 0.56%, respectively. Moreover, the prevalence of lifetime, past 12-month, and past 30-day sedative use decreased by 3.03%, 0.65%, and 0.35%, respectively. During the three waves, most trends in the prevalence of health-risk behaviors were downward, with a few exceptions: The prevalence of lifetime smoking, drinking, methamphetamine use, and sleep disturbance increased by 7.15%, 13.08%, 0.48%, and 9.06%, respectively. The prevalence of lifetime 3,4-methylene dioxy methamphetamine use (from 0.49% to 0.48%), lifetime mephedrone use (from 0.30% to 0.24%), or suicide attempts (from 2.41% to 2.46%) remained stable.

## 1. Introduction 

Adolescence is usually defined as a developmental period of immense behavioral, psychological, and social change, and also characterized as a stage of increased imitation and exploration with a range of health-risk behaviors (HRBs) [1]. To a large extent, most of the adverse health consequences experienced by adolescents are the results of their risk behaviors [2]. Although there is no uniform definition of health-risk behavior worldwide, it is generally considered as behaviors that negatively affect health, such as substance use (e.g., the use of alcohol, cigarettes, and psychoactive drugs), sleep disturbance, suicidal behaviors, and so on [3]. Those HRBs can have long lasting negative effects throughout an individual’s life, and cause a major economic and social burden to the society [4]. Adolescent HRBs have been well studied in western countries, and several national surveillance systems have also been established. The Health Behavior in School-Aged Children (HBSC) (2009/2010) showed that the prevalence of 15-year-old boys who drink alcohol at least once a week was 27% [5]; the latest 2016 National Survey on Drug Use and Health (NSDUH) in the United States report showed that the prevalence of past-year opioids use among adolescents was approximately 3.6% (corresponding to 891,000 adolescents) [6]; a previous study from the Youth Risk Behavior Surveillance System (YRBSS) suggested the distribution of sleep duration among middle school students was 5.9% for ≤4 h/weekday, 6.0% for 5 h/weekday, 17.2% for 9 h/weekday, and 10.0% for ≥10 h/weekday [7]. The report of the 2017 YRBSS demonstrated that 7.4% of high school students had attempted suicide [3]. Although there has not been a significant national surveillance system in China to monitor the HRBs among adolescents, Qu et al. found that the six-month prevalence of any mental disorder was 15.24% in 6 to 16-year-old students in Sichuan province [8]. Liu et al. reported that 19% of adolescents in Shandong province had suicidal ideation, and 7% made a suicide attempt during the past 6 months [9]. Feng et al. reported 23.6% of male and 15.3% of female middle school students had consumed alcohol in the past 30 days [10]. Our prior studies also have shown that among Chinese adolescents, the prevalence of sleep disturbance was 39.6% [11]; the prevalence of past-month opioid use and sedative use were 1.6% and 2.0%, respectively [12]; 3.1% reported having suicide attempts [13]. These results indicate that adolescent HRBs are prevalent and becoming a major public health problem, and China is no exception. However, considering the variety of public health policies and healthcare systems in different regions, the trends of adolescent HRBs may be various. A previous study in Taiwan demonstrated that the prevalence of smoking and drinking alcohol in elementary school students decreased from 2001 to 2003 [14]. A prior study using the data of YRBSS showed that from 1991 to 2011, suicide attempts decreased significantly among female students [15]. A study using the data from HBSC demonstrated that there was a rising trend of sleep problems among Finnish adolescents from 1984 to 2011 [16].

China has seen rapid social and economic changes in the past three decades, and those may also have influences on the patterns of HRBs among Chinese adolescents [17]. Yet, there is scarcely direct information on the trends in the prevalence of HRBs in Chinese adolescents. Monitoring trends in HRBs among Chinese adolescents may have implications for providing insights into public health concerns, and the information would help provide suitable recommendations and prevention strategies to schools, parents, policymakers, and stakeholders. Guangdong province is a forerunner of China’s reform and opening-up policy, with a higher speed of economic growth, resulting in adolescents being more easily exposed to HRBs. Therefore, in this study, we used data from three waves of Guangdong part of the 2012 to 2017 School-based Chinese Adolescents Health Survey (SCAHS) to estimate the prevalence of the HRBs among adolescents in Guangdong province and to test the trends in HRBs without and with adjusting for sociodemographic characteristics.

## 2. Materials and Methods

### 2.1. Study Design and Participants

Data were drawn from the SCAHS, which is an ongoing school-based study about the HRBs among Chinese adolescents (7th to 12th grade), and conducts large-scale cross-sectional surveys every two years commencing in 2007 [18,19,20]. To protect the privacy of the students, SCAHS used anonymous self-report questionnaires, which are filled out by students during class and administered by research assistants in the classrooms without the presence of teachers (to avoid any potential information bias). This study analyzed the data from three waves of the SCAHS in Guangdong Province from 2012 through 2017. Guangdong province is located in the south of mainland China, and the SCAHS conducted the survey in this province from 2012 through 2017. Overall, 71,083 students’ questionnaires were completed and qualified for this study, and the mean weighted response rate was 90.3%.

### 2.2. Ethical Statement

Written informed consents were obtained from each participating student who was at least 18 years of age. If the student was under 18 years of age, written informed consent was obtained from one of the student’s parents (or legal guardians). The study was approved by the Sun Yat-Sen University School of Public Health Institutional Review Board.

### 2.3. Measurement

The SCAHS collected data on lifetime, past 12-month, and past 30-day use of non-medical use of prescription drug (NMUPD); lifetime use of illicit drugs; lifetime use of cigarette and alcohol; past 12-month suicide attempts; past 30-day sleep disturbance. In the SCAHS, the measures of NMUPD included lifetime, past 12-month, and past 30-day nonmedical use of opioids and sedatives (coded 0 = No and 1 = Yes). The list of prescription drugs was developed with a focus on medications that were reported to be widely used by adolescent drug abusers in rehabilitation centers of China. Opioids included compounded cough syrup with codeine (codeine), compounded licorice tablets (opium), tramadol hydrochloride, and diphenoxylate. Sedatives included compounded aminopyrine phenacetin tablets (barbiturates), diazepam or triazolam (benzodiazepines), and scopolamine hydrobromide tablets (barbiturates). 

The measures of illicit drug use included lifetime use of four types of illicit drugs (coded 0 = No and 1 = Yes). The list of synthetic illicit drugs was provided by the Centre for ADR Monitoring of Guangdong, including “3,4-methylene dioxy methamphetamine (MDMA)”, “methamphetamine”, “ketamine”, and “mephedrone”.

Lifetime smoking and drinking were evaluated by asking students the question: “Have you ever smoked a cigarette (or drunk alcohol)?” (responses were coded as 0 = No and 1 = Yes). 

Past 12-month suicide attempts were assessed by the question: “During the past 12 months, how many times did you actually attempt suicide?” (responses were coded as 0 = Never and 1 = Once or more).

Past 30-day sleep disturbance was measured by the validated and extensively used the Pittsburgh Sleep Quality Index (PSQI) in Chinese [11]. The global PSQI scores range from 0 to 21 points in which higher scores indicate worse sleep quality, and a PSQI global score of above 7 points indicates poor sleep quality collectively known as *sleep disturbance*. [21].

Sociodemographic characteristics included age, gender (1 = male and 2 = female), living arrangement (1 = living with both parents, 2 = living with a single parent, and 3 = living with others), household socioeconomic status (HSS; 1 = excellent or very good, 2 = good, and 3 = fair or poor), classmate relations and teacher-classmate relations (1 = good, 2 = average, and 3 = poor), and year. 

### 2.4. Statistical Analysis

The prevalence estimates and logistic regression analyses used appropriate sampling weights and estimation procedures that accounted for the complex sampling design [22]. First, descriptive analyses were conducted to estimate the prevalence of lifetime, past 12-month, and past 30-day use of NMUPD; lifetime use of illicit drugs; lifetime use of cigarette and alcohol; past 12-month suicide attempts; past 30-day sleep disturbance. Trends in the prevalence were analyzed by using the weighted univariable and multivariable logistic regression models. A Bonferroni correction was utilized to reduce chances of type I errors of testing overall trends. Multicollinearity (using variance inflation factors) and potential interaction effects between examined factors were not identified in the final multivariable models. 

## 3. Results

Based on the data of the 71,083 sampled students (13 to 20 years old) retrieved from the SCAHS during the first wave (2012–2013) through the third wave (2016–2017), the prevalence of lifetime opioid use decreased by 4.19% (from 6.18% to 1.99%, *P*_adjusted trend_ < 0.001), the prevalence of past 12-month opioid use modestly decreased by 0.63% (from 1.74% to 1.11%, *P*_adjusted trend_ < 0.001), and the prevalence of past 30-day opioid use modestly decreased by 0.56% (from 1.08% to 0.52%, *P*_adjusted trend_ < 0.001). Moreover, the prevalence of lifetime sedative use decreased by 3.03% (from 4.05% to 1.02%, *P*_adjusted trend_ < 0.001), the prevalence of past 12-month sedative use modestly decreased by 0.65% (from 1.37% to 0.72%, *P*_adjusted trend_ < 0.001), and the prevalence of past 30-day sedative use modestly decreased by 0.35% (from 0.77% to 0.42%, *P*_adjusted trend_ < 0.001). Regarding the illicit drug use, the prevalence of ketamine use modestly decreased by 0.11% (from 0.65% to 0.54%, *P*_adjusted trend_ < 0.001). During the three waves, most trends in the prevalence rates of HRBs were downward, with a few exceptions: The prevalence of lifetime smoking increased by 7.15% (from 5.24% to 12.39%), the prevalence of lifetime drinking increased by 9.05% (from 20.10% to 29.15%), the prevalence of lifetime methamphetamine use increased by 0.48% (from 0 to 0.48%), and the prevalence of sleep disturbance increased by 9.06% (from 43.67% to 52.73%). Additionally, the prevalence of lifetime MDMA use (from 0.49% to 0.48%), lifetime Mephedrone use (from 0.30% to 0.24%), or suicide attempts (from 2.41% to 2.46%) showed a steady change during the three waves. (Table 1 and Figure 1)

As shown in Table 2, without adjusting for sociodemographic characteristics, the prevalence of lifetime and past 30-day use of opioids, lifetime mephedrone use, or lifetime ketamine use among adolescents steadily decreased during the three waves (*p* < 0.05). Most of the downward trends remained statistically significant after adjusting for age, gender, living arrangement, HSS, classmate relations, and teacher–classmate relations (adjusted trend, *p* < 0.05). However, the downward trend of lifetime ketamine use was no longer significant after adjusting for sociodemographic characteristics. Moreover, the prevalence rates of lifetime smoking, lifetime drinking, and suicide attempts steadily increased during the three waves, and these upward trends remained significant after adjusting for sociodemographic characteristics (adjusted trend, *p* < 0.05).Additionally, the prevalence of the past 12-month opioid use, lifetime sedative use, past 12-month sedative use, or past 30-day sedative use increased first in the second wave (2014–2015) and then decreased in the third wave (2016–2017). The fluctuant trends remained statistically significant after adjusting for age, gender, living arrangement, HSS, classmate relations, and teacher-–classmate relations (adjusted trend, *p* < 0.05). The trends of lifetime MDMA use and sleep disturbance decreased first in the second wave, and then increased in the third wave; these fluctuant trends remained significant after adjusting for sociodemographic characteristics (adjusted trend, *p* < 0.05).

## 4. Discussion and Implication

The present study found that the prevalence of lifetime, past 12-month, and past 30-day opioid use among adolescents decreased from the first wave (2012–2013) to the third wave (2016–2017). The prevalence of lifetime, past 12-month, and past 30-day sedative use among adolescents also decreased during the three waves, and the prevalence of ketamine use decreased from 0.65% to 0.54% during the three waves. Additionally, the prevalence of lifetime smoking and drinking increased during the three waves, the prevalence of lifetime methamphetamine use increased by 0.48%, and the prevalence of sleep disturbance increased by 9.06%. Moreover, the prevalence of lifetime MDMA use, lifetime Mephedrone use, or suicide attempts remained steady change during the three waves. These findings are an important addition to the existing literature about Chinese adolescent HRBs. Similarly, prior evidence has shown that the condition of health care for children and young people in China is improving [23].

Our multivariable binary logistic regression models demonstrated that after adjusting for sociodemographic characteristics, the downward trends were found in lifetime and past 30-day use of opioid and lifetime mephedrone use during the three waves. The fluctuant trends (increasing first in the second wave, and then decreasing in the third wave) were found in past 12-month opioid use, lifetime sedative use, past 12-month sedative use, and past 30-day sedative use among Chinese adolescents. Similarly, a recent study using data from the NSDUH suggested a steady decline in the nonmedical use of prescription opioids between 2007 and 2014 [24]. By contrast, a recent study using data from the 2004 and 2014 Spanish State Survey on Drug Use demonstrated that the prevalence of sedative misuse among school population increased significantly [25]. These different results might be related to access to these psychoactive drugs, which might vary in different countries. Some national surveillance systems, including Monitoring the Future (MTF), NSDUH, and HBSC, have been conducted in the United States and some Europe countries to monitor the drug use among adolescents, and the results of the trends from these surveillance systems allow policymakers to gauge progress toward improving behavioral health [3,5,24]. Although there is no national surveillance system about HRBs among adolescents in China, the SCAHS has conducted three surveys in Guangdong province. These downward trends might be related the SCAHS using the school-based design and conducting educational campaigns directed at schools after every cycle of the survey to increase the awareness of the adverse effects of psychoactive drug use on the development of individuals.

Additionally, our findings also showed the upward trends in lifetime smoking and lifetime drinking among adolescents in Guangdong Province. However, by comparison, a previous study demonstrated that the trends showed continuing declines in cigarette use among adolescents in the United States, and considered that raising the minimum age of legal access to tobacco products (e.g., ages 21 and 25) would lead to substantial decreases in cigarette smoking [26]. A recent review using the data from MTF, NSDUH, and YRBSS indicated the similar fluctuant decreasing trend of youth binge-drinking prevalence and thought that this reduction in youth drinking over time may reflect factors such as the enactment of a minimum legal drinking age of 21 and other alcohol regulatory policies [27]. These different findings might be related to the fact that although China has signed the Framework Convention on Tobacco Control (FCTC proposed by WHO) in 2003 and ratified this treaty in 2005, there is still no national smoke-free law in China. So far, only 13 cities of China revise or formulate local smoke-free regulations in accordance with exposure to tobacco smoke [28]. Additionally, although more and more of the public are aware of the problems (e.g., liver and cardiovascular disease, mental disorders, and unintentional injuries) caused by alcohol drinking, China is still one of the largest alcohol markets. However, there have been few alcohol policies related to health promotion over the past decade, and a prior review demonstrated that current public health systems in China cannot monitor or respond well to alcohol drinking and alcohol-related problems [29]. Moreover, the present study also showed an upward trend of suicide attempts, and a fluctuant increasing trend in the lifetime MDMA use or sleep disturbance among Chinese adolescents. Similarly, a prior study demonstrated that synthetic illicit drugs (such as MDMA) have become more popular than heroin, which was previously dominant in China [30]. Moreover, regarding education inequality in China, academic stress has become a serious social problem. To compete for resources (i.e., to outperform their competitors in average student academic performance), most Chinese high schools start earlier than 07:00, keep students in classes for long hours, assign a mass of homework, and organize countless exams [31]. Earlier school start time and higher academic stress were common among Chinese high schools students [31,32], and these situations may explain the increasing trend of sleep disturbance among adolescents in the present study. Additionally, a previous study reported that although the overall rate of suicide in China has been significantly reduced in the past decade, suicidal attempts are still a current problem among adolescents [33]. A possible explanation may be that China’s previous one-child policy results in a common phenomenon for the only child to be subject to the overemphasis on academic performance, and poor academic performance can predict suicidal ideation among Chinese adolescents [34]. Another explanation may be that due to the rigid “household registration system” of China, migrated workers are not registered as “residents” in the urban areas; as a result, their children (also called the “left-behind children”) struggle to get education and health services in the urban areas, and therefore, are more likely to be involved in mental health problems [35]. Furthermore, according to the problem behavior theory, both protective (models, controls, and supports) and risk factors (models, opportunity, and vulnerability) were reported to be associated with adolescent HRBs. Protective factors can decrease the likelihood of engaging in HRBs by providing models for positive or prosocial behavior, personal or social controls against problem behavior, and an environment of support, while risk factors may increase the likelihood of engaging in HRBs by providing greater opportunity or personal vulnerability to problem behavior involvement; then the differences in the situation of HRBs among adolescents from different countries might be related to the variety in psychosocial protective and risk factors, which may also reflect societal variation and cross-national generality [36]. Moreover, family members, peers, and the larger community are also reported to play a role in an adolescent’s life [37]. Based on these findings and the problem behavior theory, we recommend the following suggestions: (1) Strengthening the regulation of the sales of substances to adolescents, such as raising the minimum age of legal access to cigarette or alcohol; (2) developing school-based curriculum to increase students’ awareness of the adverse effects of HRBs (considering adolescents spend a major portion of their waking time in school); (3) conducting community-based education to increase parents’ knowledge about the long lasting consequence of HRBs and parenting skills; (4) encouraging families to provide stable environments and daily security for the adolescents; (5) strengthening the safeguarding including child protection, health care plans to Chinese adolescents, especially for the left-behind children and adolescents; (6) providing earlier detection and treatment services to promote resilience among adolescents who have been involved in HRBs; and (7) establishing a nation-wide active surveillance system (such as the MTF, YRBSS, NSDUH or HBSC) to monitor HRBs among adolescents in China.

The present study had several limitations that are worth noting. First, the SCAHS, which is a school-based design only included school students and did not incorporate students who had dropped out of school or were absent in school on the day that the survey was administered. HRBs might be more common among those adolescents who were absent from schools. Second, due to the cross-sectional nature of the SCAHS data, no temporal or causal relations can be made. Third, considering the SCAHS used the structured self-report questionnaires to collect data, the results may be subject to recall bias, and HRBs may be underreported due to social desirability. Despite these limitations, it is the first study to estimate the trends of HRBs among Chinese adolescents in Guangdong province. Moreover, the anonymity of the questionnaires is assured in the SCAHS, and this method may have helped to ensure the data accuracy.

## 5. Conclusions

In summary, in this study of Chinese adolescents based on the SCAHS from 2012 through 2017 in Guangdong Province, estimates of the prevalence of HRBs showed a downward trend of lifetime and past 30-day use of opioid, or lifetime mephedrone use; a fluctuant declining trend (increasing first in the second wave, and then decreasing in the third wave) of past 12-month opioid use, lifetime sedative use, past 12-month sedative use, or past 30-day sedative use; an upward trend of lifetime smoking, lifetime drinking, or suicide attempts; a fluctuant increasing trend (decreasing first in the second wave, and then increasing in the third wave) of lifetime MDMA use, or sleep disturbance.

## Figures and Tables

**Figure 1 ijerph-16-01902-f001:**
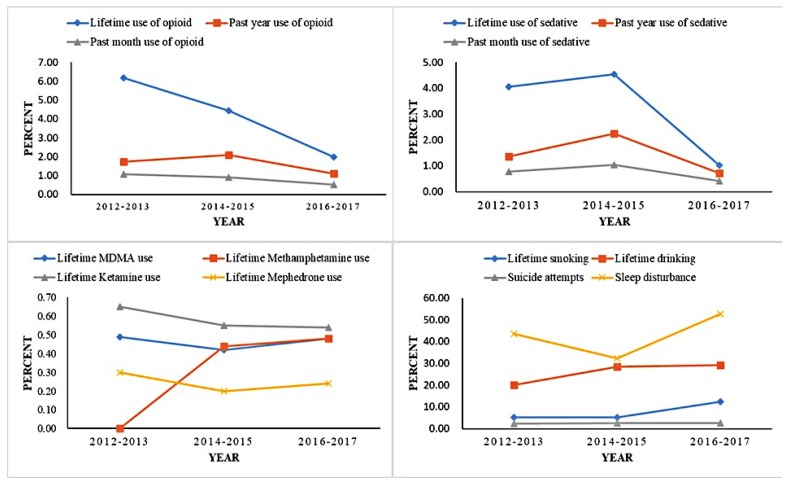
Trends in health-risk behaviors among Chinese adolescents in Guangdong Province.

**Table 1 ijerph-16-01902-t001:** Prevalence estimates of health-risk behaviors among Chinese adolescents.

Variable	Prevalence (95% CI)			Unadjusted *p* value for Trend	Adjusted *p* value for Trend *
	2012–2013	2014–2015	2016–2017		
**Opioid use**					
Lifetime	6.18 (5.87–6.49)	4.43 (4.14–4.72)	1.99 (1.81–2.17)	<0.001	<0.001
Past 12-month	1.74 (1.58–1.90)	2.09 (1.89–2.29)	1.11 (0.97–1.25)	<0.001	<0.001
Past 30-day	1.08 (0.96–1.20)	0.92 (0.78–1.06)	0.52 (0.42–0.62)	<0.001	<0.001
**Sedative use**					
Lifetime	4.05 (3.80–4.30)	4.55 (4.26–4.84)	1.02 (0.88–1.16)	<0.001	<0.001
Past 12-month	1.37 (1.21–1.53)	2.25 (2.03–2.47)	0.72 (0.60–0.84)	<0.001	<0.001
Past 30-day	0.77 (0.65–0.89)	1.04 (0.90–1.18)	0.42 (0.34–0.50)	<0.001	<0.001
**Lifetime MDMA use**	0.49 (0.41–0.57)	0.42 (0.32–0.52)	0.48 (0.38–0.58)	0.11	
**Lifetime Methamphetamine use**	0	0.44 (0.34–0.54)	0.48 (0.38–0.58)	<0.001	<0.001
**Lifetime ketamine use**	0.65 (0.55–0.75)	0.55 (0.45–0.65)	0.54 (0.44–0.64)	<0.001	0.858
**Lifetime Mephedrone use**	0.30 (0.22–0.38)	0.20 (0.14–0.26)	0.24 (0.18–0.30)	<0.001	<0.001
**Lifetime smoking**	5.24 (4.95–5.53)	5.25 (4.84–5.46)	12.39 (11.94–12.84)	<0.001	<0.001
**Lifetime drinking**	20.10 (19.57–20.63)	28.36 (27.56–29.16)	29.15 (28.29–30.01)	<0.001	<0.001
**Suicide attempts**	2.41 (2.21–2.61)	2.57 (2.35–2.79)	2.46 (2.26–2.66)	0.04	<0.001
**Sleep disturbance**	43.67 (43.00–44.34)	32.31 (31.62–33.00)	52.73 (52.04–53.42)	<0.001	<0.001

MDMA = 3,4-methylene dioxy methamphetamine; 95% CI = 95% confidence interval. * Adjusting for age, gender, living arrangement, household socioeconomic status, classmate relations, and teacher–classmate relations.

**Table 2 ijerph-16-01902-t002:** Trends in health-risk behaviors among Chinese adolescents: univariable and multivariable logistic regression showing time associated with health-risk behaviors.

Variable	OR (95% CI) ^a^		AOR (95% CI) ^a,^*	
	Wave 2 (2014–2015)	Wave 3 (2016–2017)	Wave 2 (2014–2015)	Wave 3 (2016–2017)
Lifetime Opioid use	0.71 (0.69–0.72)	0.31 (0.30–0.31)	0.68 (0.67–0.69)	0.30 (0.29–0.30)
Past 12-month Opioid use	1.20 (1.18–1.23)	0.63 (0.62–0.65)	1.14 (1.09–1.14)	0.58 (0.57–0.60)
Past 30-day Opioid use	0.85 (0.82–0.87)	0.48 (0.46–0.49)	0.78 (0.76–0.80)	0.43 (0.42–0.44)
Lifetime Sedative use	1.13 (1.12–1.16)	0.25 (0.24–0.25)	1.18 (1.16–1.19)	0.26 (0.26–0.27)
Past 12-month Sedative use	1.66 (1.63–1.70)	0.53 (0.51–0.54)	1.69 (1.66–1.73)	0.55 (0.53–0.57)
Past 30-day Sedative use	1.36 (1.32–1.40)	0.55 (0.53–0.57)	1.40 (1.36–1.45)	0.58 (0.55–0.60)
Lifetime MDMA use	0.84 (0.81–0.88)	1.04 (1.01–1.07)	0.94 (0.90–0.98)	1.11 (1.07–1.16)
Lifetime Methamphetamine use	NA	NA	NA	NA
Lifetime ketamine use	0.85 (0.82–0.88)	0.83 (0.80–0.86)	0.99 (0.95–1.02)	1.00 (0.97–1.04)
Lifetime Mephedrone use	0.67 (0.63–0.71)	0.78 (0.74–0.82)	0.74 (0.70–0.78)	0.90 (0.85–0.95)
Lifetime smoking	1.02 (1.01–1.04)	2.56 (2.53–2.59)	1.20 (1.18–1.21)	3.20 (3.17–3.24)
Lifetime drinking	1.62 (1.55–1.70)	1.76 (1.67–1.85)	1.59 (1.51–1.68)	1.71 (1.62–1.80)
Suicide attempts	1.02 (1.01–1.06)	1.07 (1.00–1.09)	1.13 (1.11–1.15)	1.15 (1.12–1.17)
Sleep disturbance	0.62 (0.61–0.62)	1.44 (1.43–1.45)	0.74 (0.73–0.74)	1.94 (1.93–1.95)

^a^ The category “wave 1 (2012–2013)” was treated as reference in the univariable and multivariable logistic regression analyses. OR = odds ratio; AOR = adjusted odds ratio; 95% CI = 95% confidence interval; MDMA = 3,4-methylene dioxy methamphetamine; NA = not applicable. * Adjusting for age, gender, living arrangement, household socioeconomic status, classmate relations, and teacher-classmate relations.

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
