# Peer review of "Trends in Health-Risk Behaviors among Chinese Adolescents"

_ijerph, 2019, doi:10.3390/ijerph16111902_

Round 1

Reviewer 1 Report

There is not a strong theorical framework... a good starting point is...

DiClemente RJ, Hansen W, Ponton LE. Handbook of adolescent health risk behavior. Plenum; 1996.

A more extensive revision and context is missing… (thouhg in dicussion section appear some relevant references)… a good starting point is… Shimazaki, T., Uechi, H., Bao, H., Deli, G., Lee, Y.-H., Miura, K., & Takenaka, K. Health behavior stage and the prevalence of health risk behaviors in inner Mongolian secondary school students: a cross-sectional study. Child & Youth Services, 2019. 1–16. https://doi.org/10.1080/0145935x.2018.1561265  This article has an interesting paragraph…

“The condition of health care for children and young people in China is exceptionally improving. The mortality rate for children under 5 years old has significantly decreased from 1996 to 2012 (The Lancet, 2016). However, as in many countries, noncommunicable diseases are a serious health concern among the young people in China. Reportedly, 70.1% of Chinese school-aged students failed to perform the recommended moderate–vigorous physical activity (Fan & Cao, 2017). Wu et al. (2005) showed 17.6% of 7–17 year olds were overweight and 5.6% were obese. Another study found 15.2% of primary and middle school students in Sichuan Province had a mental disorder (Qu, Jiang, Zhang, Wang, & Guo, 2015), and a study of 16–18 year olds in Shandong, a rural area of China, found 19% of the sample experienced suicide ideation and 7% reported having made a suicide attempt during the past six months (Liu, Tein, Zhao, & Sandler, 2005). A study on smoking status in Chinese children found even children as young as 14 developed a smoking habit (Hesketh, Ding, & Tomkins, 2001). A meta-analysis on alcohol consumption in Chinese adolescents reported that 23.6% of male and 15.3% of female middle school students had consumed alcohol within 30 days before being questioned (Feng & Newman, 2016). The United Nations International Children’s Emergency Fund (2015) reported health disparities between children in urban and those in rural areas in China were significant health problems”

There is also a lack of a complete data sheet survey, the website link of the School-based study Chinese Adolescents Health Survey (if it would exist...), the complete or partially annexed questionnaire... 

In order to improve their "Discussion and Implication", I suggest them review this book and other about prevention and treatment…

Gullotta, Thomas P.  Plant, Robert W. Evans, Melanie A. Handbook of Adolescent Behavioral Problems: Evidence-Based Approaches to... Springer; 2014.

and this papers and others about comparative research

Jessor R, Turbin MS, Costa FM, Dong Q, Zhang H, Wang C. Adolescent problem behavior in China and the United States: A cross-national study of psychosocial protective factors. Journal of Research on Adolescence. 2003;

Author Response

Response to Reviewer 1 Comments

Point 1: There is not a strong theorical framework... a good starting point is...

DiClemente RJ, Hansen W, Ponton LE. Handbook of adolescent health risk behavior. Plenum; 1996.

Response 1: We truly appreciate your time for reviewing our article. According to your suggestion, we have revised the Introduction section and cited this reference (please see lines 31-32).

Point 2: A more extensive revision and context is missing… (thouhg in dicussion section appear some relevant references)… a good starting point is… Shimazaki, T., Uechi, H., Bao, H., Deli, G., Lee, Y.-H., Miura, K., & Takenaka, K. Health behavior stage and the prevalence of health risk behaviors in inner Mongolian secondary school students: a cross-sectional study. Child & Youth Services, 2019. 1–16. https://doi.org/10.1080/0145935x.2018.1561265   This article has an interesting paragraph…

“The condition of health care for children and young people in China is exceptionally improving. The mortality rate for children under 5 years old has significantly decreased from 1996 to 2012 (The Lancet, 2016). However, as in many countries, noncommunicable diseases are a serious health concern among the young people in China. Reportedly, 70.1% of Chinese school-aged students failed to perform the recommended moderate–vigorous physical activity (Fan & Cao, 2017). Wu et al. (2005) showed 17.6% of 7–17 year olds were overweight and 5.6% were obese. Another study found 15.2% of primary and middle school students in Sichuan Province had a mental disorder (Qu, Jiang, Zhang, Wang, & Guo, 2015), and a study of 16–18 year olds in Shandong, a rural area of China, found 19% of the sample experienced suicide ideation and 7% reported having made a suicide attempt during the past six months (Liu, Tein, Zhao, & Sandler, 2005). A study on smoking status in Chinese children found even children as young as 14 developed a smoking habit (Hesketh, Ding, & Tomkins, 2001). A meta-analysis on alcohol consumption in Chinese adolescents reported that 23.6% of male and 15.3% of female middle school students had consumed alcohol within 30 days before being questioned (Feng & Newman, 2016). The United Nations International Children’s Emergency Fund (2015) reported health disparities between children in urban and those in rural areas in China were significant health problems”.

Response 2: Thank you for your kind suggestion. We have cited Shimazaki’s study to the Discussion section (please see lines 187-188). Considering the discussion focused on the trends in the health-risk behaviors, the studies about the prevalence rates in the paragraph your mentioned above have also been added to our Introduction section in order to expand the existed studies about the prevalence rates of health-risk behaviors among Chinese adolescents (please see lines 48-52).

Point 3: There is also a lack of a complete data sheet survey, the website link of the School-based study Chinese Adolescents Health Survey (if it would exist...), the complete or partially annexed questionnaire... 

Response 3: Thank you for your kind suggestion. However, we apologize for not having a website about the School-based study Chinese Adolescents Health Survey (SCAHS) now. We have published a series of papers using the data from the SCAHS [1-5], we would like to establish a website for the SCAHS in the future.

Point 4: In order to improve their "Discussion and Implication", I suggest them review this book and other about prevention and treatment…

Gullotta, Thomas P.  Plant, Robert W. Evans, Melanie A. Handbook of Adolescent Behavioral Problems: Evidence-Based Approaches to... Springer; 2014.

and this papers and others about comparative research

Jessor R, Turbin MS, Costa FM, Dong Q, Zhang H, Wang C. Adolescent problem behavior in China and the United States: A cross-national study of psychosocial protective factors. Journal of Research on Adolescence. 2003;

Response 4: We truly appreciate your suggestions. As suggested, we have reviewed these two literatures and cited them to our revised manuscript (please see lines 243-252). Moreover, according to the Handbook of Adolescent Behavioral Problems, we have adjusted the recommendations to make them more clear (please see lines 256-260).

References:

1.     Guo L, Huang Y, Xu Y, Huang G, Gao X, Lei Y, Luo M, Xi C, Lu C: The mediating effects of depressive symptoms on the association of childhood maltreatment with non-medical use of prescription drugs. J Affect Disord 2018, 229:14-21.

2.     Guo L, Luo M, Wang W, Huang G, Zhang WH, Lu C: Association between weekday sleep duration and nonmedical use of prescription drug among adolescents: the role of academic performance. Eur Child Adolesc Psychiatry 2019.

3.     Guo L, Luo M, Wang W, Xiao D, Xi C, Wang T, Zhao M, Zhang WH, Lu C: Association between nonmedical use of opioids or sedatives and suicidal behavior  among Chinese adolescents: An analysis of sex differences. Aust N Z J Psychiatry 2018:1220868576.

4.     Guo L, Wang W, Gao X, Huang G, Li P, Lu C: Associations of Childhood Maltreatment with Single and Multiple Suicide Attempts  among Older Chinese Adolescents. J Pediatr 2018, 196:244-250.

5.     Guo L, Xu Y, Deng J, Gao X, Huang G, Huang J, Deng X, Zhang WH, Lu C: Associations between childhood maltreatment and non-medical use of prescription drugs among Chinese adolescents. ADDICTION 2017, 112(9):1600-1609.

Reviewer 2 Report

This is a very well written paper and in particular, on a very interesting area and a group of population.

Author Response

Response to Reviewer 2 Comments

Point 1: This is a very well written paper and in particular, on a very interesting area and a group of population.

Response 1: Thank you for carefully and patiently reviewing our manuscript. We apologize for the grammatical errors in the original manuscript (Pdf document), and we have revised these errors according to your suggestions.

Moreover, regarding the question about past 12-month suicide attempts, we have demonstrated that past 12-month suicide attempts were assessed by the question “During the past 12 months, how many times did you actually attempt suicide?”. This item has been utilized in our previous publications [1, 2]. Although our prior study also found that there might be differences in adolescents with single and multiple suicide attempts, the present study focused on adolescents having suicide attempts or not, and utilized the dichotomized variable of suicide attempts to measure the prevalence of suicide attempts.

References:

1.     Guo L, Xu Y, Deng J, Huang J, Huang G, Gao X, Li P, Wu H, Pan S, Zhang WH et al: Association between sleep duration, suicidal ideation, and suicidal attempts among Chinese adolescents: The moderating role of depressive symptoms. J Affect Disord 2017, 208:355-362.

2.     Guo L, Xu Y, Deng J, Huang J, Huang G, Gao X, Wu H, Pan S, Zhang WH, Lu C: Association Between Nonmedical Use of Prescription Drugs and Suicidal Behavior Among Adolescents. JAMA PEDIATR 2016, 170(10):971-978.
